# Surface Modification of Carbon Microspheres with Guanidine Phosphate and Its Application as a Flame Retardant in PET

**DOI:** 10.3390/polym12081689

**Published:** 2020-07-29

**Authors:** Shan Jiang, Cheng Ji, Dan Zha, Yonghong Ding, Dun Wu, Qiang Yu

**Affiliations:** School of Materials Science and Engineering, Jiangsu Collaborative Innovation Center for Photovolatic Science and Engineering, Changzhou University, Changzhou 213614, China; 18000151@smail.cczu.edu.cn (C.J.); zhadan@toray-tpn.cn (D.Z.); dyh@cczu.edu.cn (Y.D.); wudun@cczu.edu.cn (D.W.); yuqiang@cczu.edu.cn (Q.Y.)

**Keywords:** PET, carbon microspheres, flame retardant, guanidine phosphate, surface modification

## Abstract

Composites based on polyethylene terephthalate (PET) and surface-modified carbon microspheres (CMSs) were prepared by melt mixing. The surface modification of CMSs was conducted with low-temperature plasma technology first, and a phosphorus-nitrogen flame retardant, guanidine phosphate (GDP), was then grafted onto CMSs. The modification of CMSs was done to improve both the filler dispersity in the PET matrix and the flame-retardant performance of composites. The obtained CMSs-GDP was characterized by FTIR spectra and a scanning electron microscope (SEM). The grafting ratio was characterized and calculated by thermal gravimetric analysis (TGA). The grain size analysis was used to describe the dispersity of CMSs. The mechanical properties of the PET/CMSs-GDP composite were measured using a universal testing machine. The PET/CMSs-GDP composite can achieve a limiting oxygen index (LOI) value of 32.4% and a vertical burning test (UL94) V-0 rating at 3% CMSs-GDP loading.

## 1. Introduction

Flame retardants for polyethylene terephthalate (PET) have been a constant research hotspot. The commonly reported flame retardants used with PET can be divided into various kinds, such as traditional halogen-containing flame retardants, intumescent flame retardants, and inorganic nanometer fire retardants [1]. However, with recent stricter requirements for safety and environmental protection, the development of halogen flame retardants has been limited [2]. Many new inorganic nanometer halogen-free and intumescent flame retardants have been synthesized [3]. For example, Cai [4] studied the effects of laminated metal lanthanum phenyl phosphonate (LaPP) on the thermal stability, flame retardation characteristics, and mechanical properties of a PET/microcapsule red phosphorus system (PET/GF-MRP) reinforced by glass fiber, whose LOI can reach 28.9%. Wu [5] used liquid bisphenol A, double (diphenyl phosphorous) (BDP) to pre-disperse, so that carbon nanotubes could be dispersed uniformly in PET by melting and blending, which can enhance the flame-retardant behavior. Feng [6] incorporated a flame-retardant system consisting of 1-hydroxymethyl-1, 1-diphosphate (HEDP) and ammonium sulfamate into polyester fabric for flame retardation; the LOI was increased to 28%, and the results showed that the PET composite material can achieve the UL94-V0 level.

Carbon microspheres are a new type of nano-inorganic carbon material that have received much attention recently [7]. CMSs have been widely studied because of their excellent heat resistance, chemical stability, and good electrical and thermal conductivity [8]. However, CMSs are bonded by means of a strong π–π conjugated system and van der Waals forces, so their particulates are markedly clustered and their dispersion in matrix material is poor [9]. Therefore, the modification of dispersion and interface compatibility of CMSs has received emphasis at this stage. The most common modification method of CMSs is chemical pretreatment with sulfuric acid (H_2_SO_4_) and nitric acid (HNO_3_) to modify the surface of carbon microspheres [10,11]. Qi [12] used sulfuric acid and nitric acid to modify carbon spheres, and their dispersion and interface compatibility were improved. Eon [13] synthesized a sulfur/graphitic hollow carbon sphere to improve the surface of carbon spheres; the carbon spheres were prepared from pyrolysis of a homogenous mixture of monodispersed spherical silica, and their dispersion was modified significantly. Recently, the application of CMSs in flame-retardant material has received attention [14]. CMSs can expand when heated, making them a new additive intumescent flame retardant [14]. When a composite material is burned, the CMSs can expanded to a porous structure, which can obstruct the combustible gas; thus, they have flame-resistant effects [15,16]. For example, Yaru [17] used MH@CMSs to microencapsulate PET to obtain microencapsulated CMSs coated by magnesium hydroxide flame retardants; the LOI of the composite material was enhanced to 27.4%. Niu [7] added CMSs coated with magnesium hydroxide (Mg(OH)_2_ @CMSs) to PET to improve the flame retardancy of PET, with an LOI that reached 27.5%.

The aim of the present work is to synthesize a graft-modified CMSs flame retardant, called CMSs-GDP, with high flame retardant efficiency and good mechanical properties. The chemical structure of CMSs-GDP was characterized by FTIR, XRD, and TGA. The dispersion and interface compatibility were characterized by grain size analysis and SEM. The flame retardant and mechanical properties were comprehensively investigated.

## 2. Experimental

### 2.1. Material and Instruments

Beta-cyclodextrin and guanidine phosphate were purchased from Aladdin Chemical Reagent Corp. (Shanghai, China). Carbon microspheres were synthesized from beta-cyclodextrin by hydrothermal synthesis. PET resin (CR-8663) was purchased from Changzhou Huarun Chemical Holdings Co., Ltd. (Changzhou, China). Other chemicals were purchased from China National Pharmaceutical Group (Shanghai, China). The low-temperature plasma processor (WAOL 2000-Cr) was acquired from Suzhou Qianxun Electronics Co., Ltd. (Suzhou, China).

The synthesis and Preparation of CMSs, CMSs-PLS, and CMSs-GDP is shown in Figure 1.

Carbon microspheres were compounded by the hydrothermal synthesis of cyclodextrin. The hydrothermal synthesis temperature and time were 160 °C and 10 h, respectively. Low-temperature plasma technology was adopted to modify the surface of CMSs, giving birth to CMSs-PLS; the low-temperature plasma time is set to 300 s and the power is 250 W. A phosphorus-nitrogen flame retardant, guanidine phosphate (GDP), was grafted onto CMSs by impregnating and grafting (see Figure 1) to improve the flame retardant characteristics, interface compatibility, and dispersity of CMSs; the resulting material was named CMSs-GDP. The main chemical reaction is the amidation of the amine group and the alkyl group on the CMS.

### 2.2. Preparation of Composite Material

PET/CMSs-GDP composites were prepared using an Internal Mixer (model PPT-3/ZZL-40, Mester Industrial System, Shanghai, China) rotating at 40 r/min and a mixing chamber with the capacity to hold 45 g of material. Composites with 0, 1, 2, and 3 wt % of CMSs-GDP were prepared. The mixing temperature was monitored using a thermocouple and reached 260 °C in 5 min of mixing. The compositions were named according to the content of CMSs-GDP, for example, the composite with the addition of 1 wt % of CMSs-GDP is referred to as PET/1%CMSs-GDP.

### 2.3. Characterization of Composite Material

The thermal properties of the composites were determined by thermal gravimetric analysis (TGA). The TGA equipment used was a TA Instruments QS100 calorimeter (TA Instruments, New Castle, state abbreviation, USA), and the analysis was conducted under N_2_ atmosphere. The PET/CMSs-GDP composite was analyzed according to the following procedures; samples were heated to 800 °C at 10 °C min^−1^.

TGA-FTIR was carried out to characterize the flame-retardant mechanism of composite material.

To analyze the fracture surface morphology of the composites by SEM, the sample was fracture in liquid nitrogen and were cryogenically fractured and coated with a thin layer of gold. The fracture surface was observed under an SEM FEI Inspect S50 (FEI, Portland, OR, USA), operating at 15 keV.

The vertical burning was tested by UL-94 flame chamber (Dongguan City Star Joe Equipment Co., Ltd., Dongguan, China) according to ASTMD 3801-2010 standard. The samples were processed in a size of 130 mm × 10 mm × 3 mm and fixed vertically above a cotton patch during tests.

The charred residue structure was obtained by the Hitachi X650 scanning electron microscopy (SEM, Hitachi Limited, Tokyo, Japan). Moreover, the combustion performance was measured using the HC-2 oxygen index meter (Dongguan Toyo Machinery Co., Ltd., Dongguan, China), according to the ASTM D2863 standard procedures.

The tensile properties were measured using a WD20D electronic universal tester (Jintan Medical Instrument Factory, Changzhou, China) according to ISO527 standard. The samples were processed in a size of 25 mm × 4 mm × 2 mm, and the crosshead speed is set to 400 mm min^−1^.

## 3. Results and Discussion

### 3.1. Characterization of CMSs

FTIR was used to investigate the chemical structure of carboxyl and hydroxyl groups in CMSs. It can be seen in Figure 2 that the absorption band of carboxyls and hydroxyls is evident for CMSs-PLS. From Figure 2, the characteristic absorption band at 3390 cm^−1^ is the stretching vibration peak of hydroxyl. The characteristic absorption band at 1662 cm^−1^ is the stretching vibration peak of carboxyl, and the characteristic absorption band at 1031 cm^−1^ is the bending vibration peak of carboxyl [18]. Thus, carboxyl and hydroxyl were proved to be grafted onto CMSs by means of low-temperature plasma technology. Figure 2 clearly shows that the GDP was chemically grafted onto the CMSs by impregnation. Some characteristic absorption bands were present in the FTIR spectrum of graft-modified CMSs. The characteristic absorption band at 3411 cm^−1^ is the N–H stretching vibration peak of amide bonds. The characteristic absorption band at 1666 cm^−1^ is the C=O in-plane flexural vibration peak of amide bonds. The characteristic absorption band at 1093 cm^−1^ is the P–O stretching vibration peak of the phosphate group.

Raman spectroscopy can be used to characterize the structural defects of CMSs. In Figure 3, which shows the Raman spectra of modified CMSs, there are two characteristic absorption bands. The characteristic absorption band at 1583 is the characteristic peak of sp^2^ carbon atoms, called the G peak, and the characteristic absorption band at 1374 is the characteristic peak of defects of carbon atoms, called the D peak [19]. The existence of the D peak (Figure 3) demonstrates that the original sp^2^ hybridization of the system was damaged. This result suggests that carboxyl and hydroxyl groups were grafted onto the CMSs by the low-temperature plasma treatment, which can destroy the sp^2^ hybridization of the original system. The I(D)/I(G), the peak ratio between the D peak and the G peak, represents the relationship between the two characteristic peaks [20]. A greater value of I(D)/I(G) denotes more crystal defects in the carbon atom. The I(D)/I(G) of CMSs-GDP was 0.54, more than the value of CMSs-PLS and CMSs, which represents that GDP was grafted on the surface of CMSs.

Grain size analysis was used to provide the degree of dispersion of CMSs. The polydispersity coefficient (PDI) (Table 1) is a key parameter of grain size analysis that can be used to characterize the dispersion of a material [21]. A lower PDI means the stability and the dispersity of a material are better. On the contrary, a higher PDI means the stability and dispersity of a material are poor. The grain size of CMSs-PLS with discharge power of 250 W and processing time of 5 min was 343 nm, which was suitable and uniformitarian. However, the same batch of CMSs-PLS shows different grain size; this was caused by different discharge power and time. When the discharge power is lower than 250 W, the CMSs cannot be completely modified, which can lead to the reunion of CMSs, so the grain size was larger. When the discharge power is higher than 250 W, the structure of CMSs was broken into char residue, which cannot be dispersed in solution, so the grain size was larger [22]. Therefore, as suggested by the results in Table 1, a discharge power of 250 W and processing time of 5 min led to the best chemical modification of CMSs-PLS.

Figure 4 and Figure 5 provide further evidence that the GDP was chemically grafted onto the CMSs. Thermogravimetric analysis (TGA) was utilized to calculate the grafting ratio of the GDP onto CMSs [23]. The grafting ratio can be calculated by means of the difference in the weight losses at the various characteristic peaks. There were two characteristic peaks on the derivative thermogravimetry (DTG) curve of CMSs (Figure 4 and Figure 5). Generally, the peak at 60 °C is the water peak, which is generated by the moisture loss from raw CMSs. The peak at 429 °C is the weight loss peak of CMSs. However, in Figure 5, in addition to the two typical peaks, a new peak at approximately from 270 °C to 350 °C can be seen; these are the weight loss peaks of grafted GDP, and it proves that the GDP unit was chemically grafted onto the CMSs and the grafting ratio of CMSs-GDP can reach to 10.73%. In conclusion, the GDP was successfully grafted onto the surface of CMSs.

### 3.2. Flammability of PET and Composite Materials

In Table 2, t1 represents the residual flame time of first combustion, t2 represents the residual flame time of second combustion, and t3 represents the flame spark resident time of second combustion. It is easily shown that the flame retardant behavior of pure PET is poor, and its rating is the HB, according to the UL94-V test. After the addition of CMSs-GDP, t1 and t2 are both less than 10 s, t1+t2 is less than 30 s, and t2+t3 is less than 10 s, which all conform to the UL94-V0 rating. Simultaneously, the samples all underwent incomplete combustion, and the droppings did not ignite cotton wool. As a result, the composite material achieves the UL94-V0 rating, with flame-retardant behavior that is markedly better than that of pure PET. With an increase in the amount of CMSs, the flame-retardant behavior of the composite material is much more effective.

As a kind of flame retardant, CMSs can form a substrate surface protective layer during combustion, which has the effect of heat insulation and oxygen insulation. Meanwhile, flammable gas and heat transfer were also prevented by CMSs, which can interrupt combustion and enhance the flame retardant behavior of the composite material.

As an intumescent flame retardant, three elements must be provided: the carbon source, the air source, and the acid source. CMSs with hydroxy groups can provide the carbon source, founding a compact carbon layer, making heat difficult to penetrate the condensed phase, preventing oxygen from entering the combustion zone, and preventing the degradation of the gas on the surface of the material. The phosphorus-nitrogen flame retardant GDP can provide the acid source and air source: GDP combined with PET will thermally decompose phosphoric acid, phosphoric anhydride, and other substances during combustion. Phosphoric acid will be further dehydrated to form metaphosphoric acid and pyrophosphoric acid, which will produce oxygen-containing polymer to form a dense carbon layer during PET decomposition. GDP combined with PET will thermally decompose non-flammable gases such as NH_3_ and NO_2_, which can dilute combustible gas, and play a flame-retardant role in the gas phase. Simultaneously, CMSs become an inorganic residue by expanding when heated. Thus, the flame retardant characteristics of the matrix material can be improved.

### 3.3. Thermal Properties and Degradation Mechanism of PET Composite Material

Figure 6 and Figure 7 show the TGA curves in nitrogen of pure PET and PET/CMSs-GDP composites. The TG and DTG data are listed in Table 3 including the initial decomposition temperature (T_5%_ was the temperature of 5 wt % weight loss), temperature of maximum weight loss(T_max_), the rate of Tmax and the residue of composites at 800 °C.

Figure 6a and Figure 7a show that the PET begins to decompose at a temperature (T5%) of ~420 °C, the maximum decomposition temperature Tmax is 464 °C, and the maximum thermal weight loss rate is 39.8 wt % min^−1^. Figure 6a and Figure 7a show that when adding 1% of CMSs, the T_5%_ of the composite increased to 422 °C, and the maximum thermal weight loss rate decreased to 38.2 wt % min^−1^. As the amount of carbon microspheres increased, the initial decomposition temperature could be increased to 426 °C, and the maximum weight loss temperature increased to 468 °C. At the corresponding maximum weight loss rate was reduced to 36.4 wt % min^−1^.

As a result, the addition of carbon microspheres can increase the degradation temperature of PET and reduce the rate of degradation when carbon microspheres were burned, it can be expanded and broken to produce some acidic material and inorganic carbon layer on the surface of PET, which can lead to the reduction of the heat transmission from outside and the fuel gas diffusion from inside. When CMSs-GDP was added, the decomposition temperature and maximum decomposition temperature of the PET composites were further increased, and the thermal weight loss rate was further decreased. When the CMSs-GDP addition amount reached 3%, the T_5%_ reached the highest 435 °C, the maximum decomposition temperature increased to 473 °C, and the maximum thermal weight loss rate reduced to 34.4 wt % min^−1^. Thus, the modified CMSs-GDP can further improve the thermal stability of the PET matrix material and reduce combustion. The heat release rate in the process, combined with the flame retardant experimental data, also shows that the introduction of CMSs-GDP can further improve the flame-retardant properties of PET composites.

In order to further investigate the relationship between the microstructure and flame retardant, the outer surface of thermo-oxidative decomposition char residue of samples at 800 °C in a muffle furnace for 10 min under air atmosphere were inspected by SEM, and the micrographs are shown in Figure 8. As shown in Figure 8a,b, there were many big holes and a fragile fractured surface on the pure PET which was not modified after combustion, suggesting a large amount of the heat transmission and gas diffusion. In the cases of PET/1%CMSs-GDP (see Figure 8c,d), the outer surface of the composite becomes compact and intact, and it can be observed in the structure of CMSs that the size and distribution of holes were reduced. Compared with the pure PET, the crack and the holes was significantly decreased. Carbon microspheres were expanded when heated, then CMSs was fractured to form a substrate surface protective layer during combustion. The inorganic carbon layer was covered on the surface of PET basis material, which can insulate the flammable gas and quality of heat. Due to the flame retardant nature of CMSs-GDP, the heat release rate was reduced and the flame retardant properties of PET were optimized. From Figure 8g,h, it can be observed that the structure of CMSs-GDP was significantly increased and an inorganic carbon layer formed by CMSs-GDP was covered on the surface of PET. This not only effectively protected the internal structure, but also increased the barrier closeness, leading to the reduction of heat transmission from outside and fuel gas diffusion from inside. Then, the non-flammability and high performance of PET/CMSs-GDP were exhibited.

Figure 9, Figure 10, Figure 11 and Figure 12 show the FTIR of the PET/CMSs-GDP composite at different temperature. It can be seen from Figure 9 that at the beginning of decomposition (T_5%_), the characteristic peaks of PET segments are obvious. The characteristic peaks at 2740 cm^−1^ is the expansion and contraction of C-H. The characteristic peaks at 1759 cm^−1^ is the typical stretching vibration peak of C=O. The peaks at 1263 cm^−1^ is the characteristic peak of CO–O–C, and the characteristic peak of C–O–C is at 1091 cm^−1^. The characteristic peaks at 2349 cm^−1^ and 1340 cm^−1^ represent the absorption peak of CO_2_, and the absorption peak of CO was at 2150 cm^−1^. This indicates that CO_2_ and CO are produced in the combustion process of PET and composite materials. Comparing the FTIR of the same component in Figure 9, Figure 10, Figure 11 and Figure 12, it can be clearly seen that, after the addition of 3% CMSs-GDP, the characteristic peaks of PET segments are weaker than other composites, which indicates the degradation of PET is significantly reduced.

During the decomposition process, the infrared characteristic peaks of CO_2_ (2349 cm^−1^, 1340 cm^−1^) and CO (2150 cm^−1^) of pure PET are high. With the addition of CMSs-GDP, the content of CO_2_ is increased, and the content of CO is reduced.

It is clearly demonstrated in Figure 13a that the GDP was grafted onto CMSs. The CMSs were coated with GDP, and the CMSs-GDP was incorporated into PET. The consistency of the composite material consisting of grafted CMSs with PET is homogeneous. Figure 13b shows that there are many CMSs with grafted GDP dispersed in the PET material. The dispersion of CMSs was improved, and the combination of CMSs and PET was stable and showed good performance.

### 3.4. Crystallization Performance of PET Composite Material

Figure 14 and Figure 15 and Table 4 show the DSC data for PET composites. It can be seen from the figure that the introduction of CMSs-GDP can further improve the crystallization ability and crystallization rate of PET composites. The crystallization temperature of pure PET is 164.61 °C and the crystallinity is 22.27%. With addition of 1%CMSs-PLS, the crystallization temperature is increased to 209 °C and the crystallinity is increased to 30.82%. The crystallinity increases with the addition of the amount, and the crystallinity after adding 1%CMSs-GDP is 36.43%. When 3% CMSs-GDP is added, the crystallinity of PET/3%CMSs-GDP is increased to 34.29%, so it can be concluded that the CMSs-GDP is a kind of nucleating agent when combined with PET material. It can significantly increase the crystallinity and crystallization rate of the PET matrix, making it easier to form a network crystal structure. In summary, CMSs-GDP can increase the crystallinity and crystallization rate of the PET and its effect is better than CMSs-PLS.

### 3.5. Mechanical Properties of PET Composite Material

Figure 16 shows that the tensile strength of pure PET is 47.20 MPa. With the addition of CMSs, CMSs-PLS, and CMSs-GDP, respectively, the tensile strength of the composite material shows an increasing trend on the whole. The tensile strength of the PET/CMSs-GDP composite material can reach up to 61.20 MPa. As a result, the tensile strength of these three different kinds of composites is much higher than that of pure PET. Carbon microspheres can be a nucleating agent (Figure 16), whereby the nucleation of PET is accelerated and the crystallization of PET is promoted, meaning that the chain movement of PET molecules was more difficult and the system tensile strength was significantly improved. However, with an increase in the amount of CMSs, the tensile strength showed a downward trend, and with the increasing crystallization of PET, the density of polar groups in the composite material increased and the material became more brittle. Therefore, the tensile strength of the material decreased.

Figure 17 shows that the elongation of pure PET is 16.85%. With the addition of CMSs, CMSs-PLS, and CMSs-GDP, respectively, the elongation of the composite material shows a decreasing trend on the whole. The elongation at break decreased the most after pure carbon microspheres were added. This is due to the poor dispersibility of carbon microspheres in PET, resulting in a decrease in the chain flexibility of PET. The surface of CMSs modified by low-temperature plasma has hydroxyl and carboxyl groups, so the dispersibility in PET improves, which improves the chain flexibility of PET. Because CMSs-GDP is grafted with guanidine phosphate, the increase in the particle size of the carbon microspheres hinders the chain flexibility of PET, so the elongation at break decreases.

## 4. Conclusions

In this study, a series of modified CMSs were successfully synthesized. The effect of the low-temperature plasma technology was shown to have a strong influence on the dispersity of CMSs. The obtained CMSs-PLS shows the best dispersity in polar solvent and the grain size is ~340 nm. The phosphorus-nitrogen flame retardant guanidine phosphate (GDP) was grafted on carbon microspheres (CMSs-GDP) by impregnation, and the obtained CMSs-GDP shows the best flame retardant and thermodynamic properties. The addition of CMSs-GDP as a flame retardant additive has significantly affected the flame retardant of PET composite. With an increase in the amount of CMSs, the flame retardant behavior of the composite material was much more effective. This results in a LOI value of PET/CMSs-GDP composite up to 32.4% and achieving a UL94-V0 rating. The addition of CMSs-GDP was advantageous to the decomposition of PET; it accelerated the char formation and increased the amount of char residue, resulting in the reduction of the heat transmission from the outside and the fuel gas diffusion from the inside. The non-flammability and high performance of PET/CMSs-GDP were exhibited. For the various types of modified CMSs, the tensile strength of the composite material shows an increasing trend on the whole. The tensile strength of the PET/CMSs-GDP composite material can reach up to 61.20 MPa. The addition of CMSs thus has a great influence on the mechanical properties of PET.

## Figures and Tables

**Figure 1 polymers-12-01689-f001:**
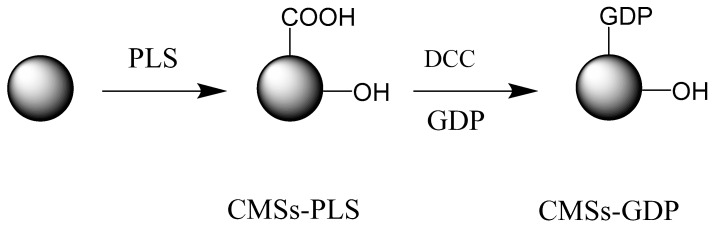
Synthesis of graft-modified carbon microspheres (DCC: Dicyclohexylcarbodiimide; GDP: C_2_H_10_N_6_•H_3_PO_4_).

**Figure 2 polymers-12-01689-f002:**
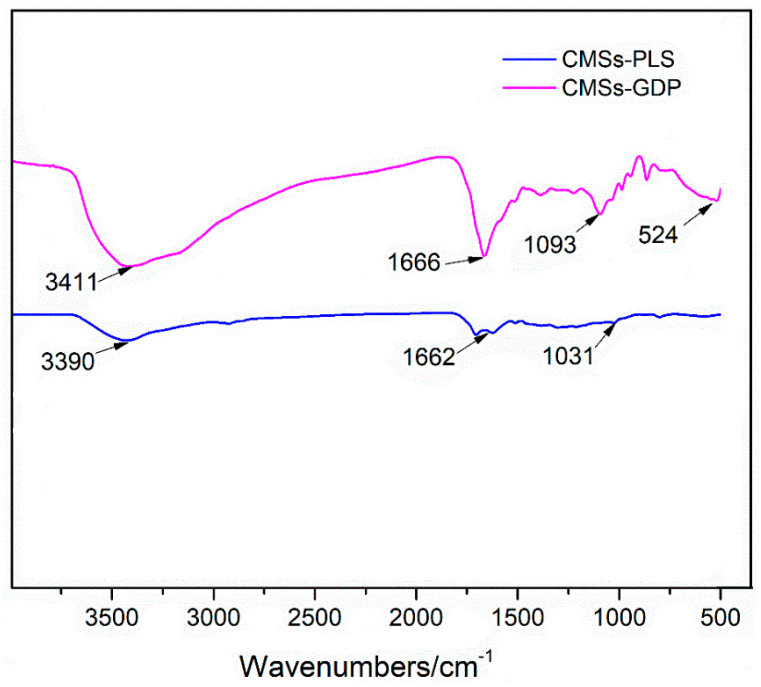
FTIR of CMSs-PLS and CMSs-GDP.

**Figure 3 polymers-12-01689-f003:**
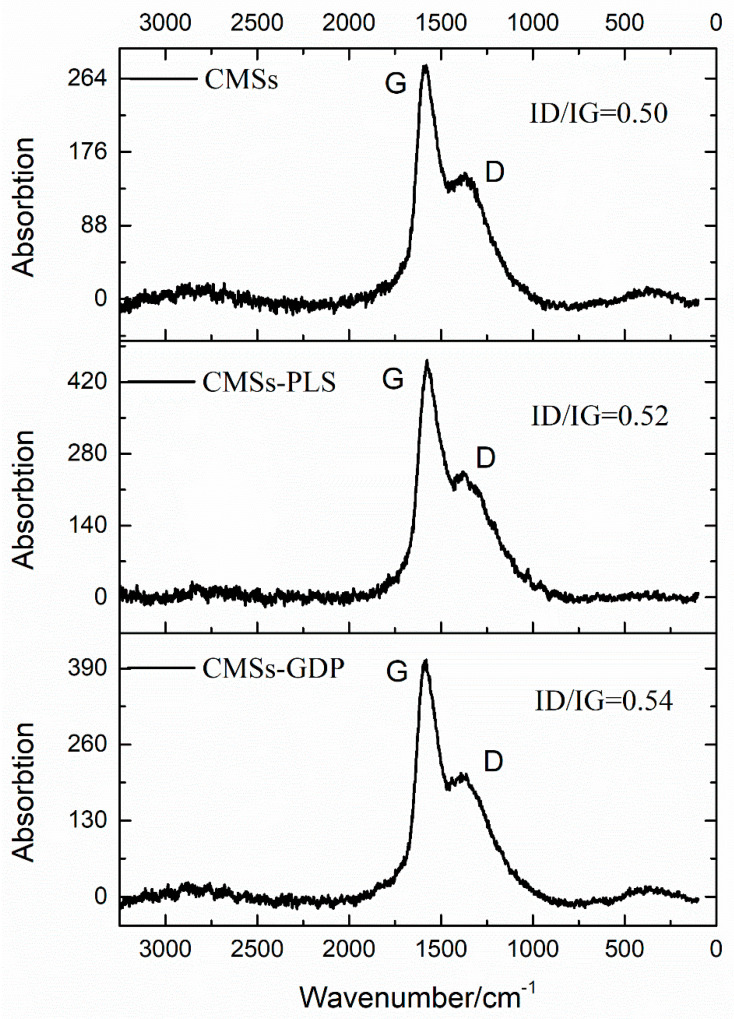
Raman spectrum of CMSs, CMSs-PLS, and CMSs-GDP.

**Figure 4 polymers-12-01689-f004:**
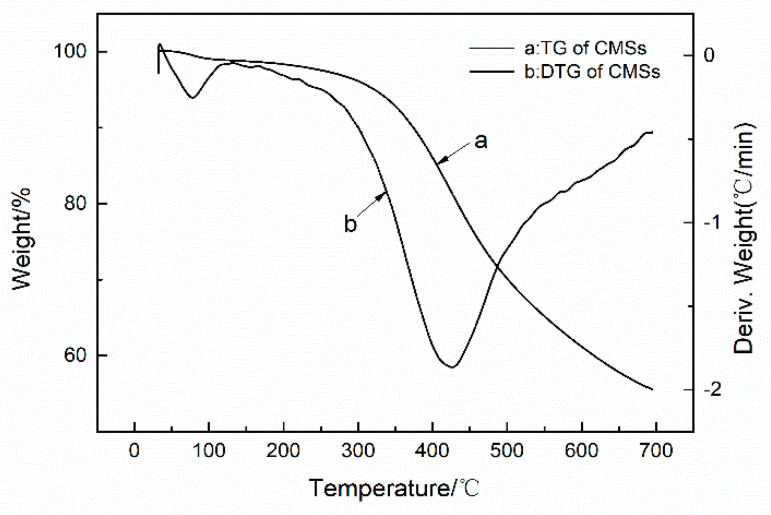
TG and DTG of carbon microspheres.

**Figure 5 polymers-12-01689-f005:**
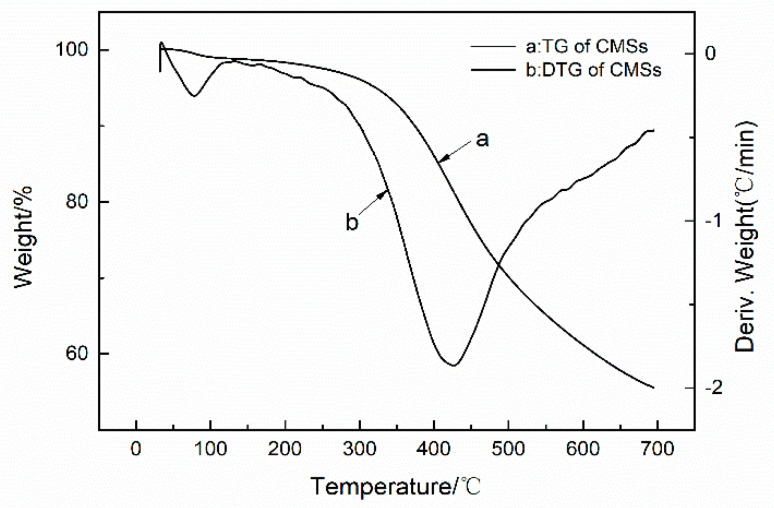
Thermal gravimetry (TG) and derivative thermogravimetry (DTG) of CMSs-GDP.

**Figure 6 polymers-12-01689-f006:**
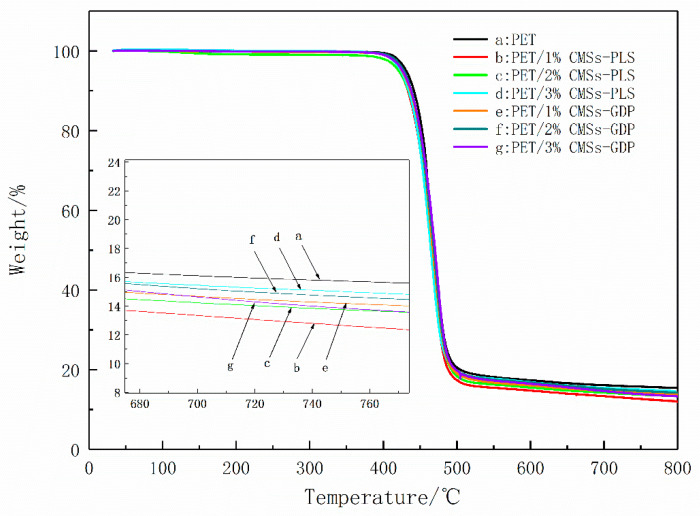
The TG curves of PET, PET/CMSs-PLS, and PET/CMSs-GDP.

**Figure 7 polymers-12-01689-f007:**
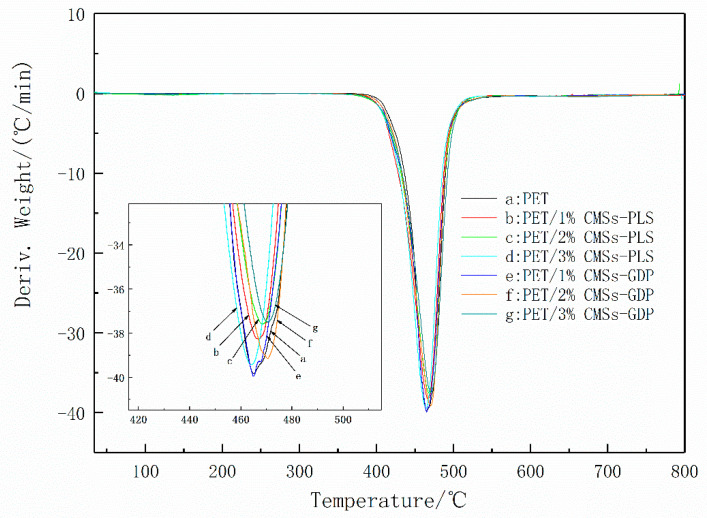
The DTG curves of PET, PET/CMSs-PLS, and PET/CMSs-GDP.

**Figure 8 polymers-12-01689-f008:**
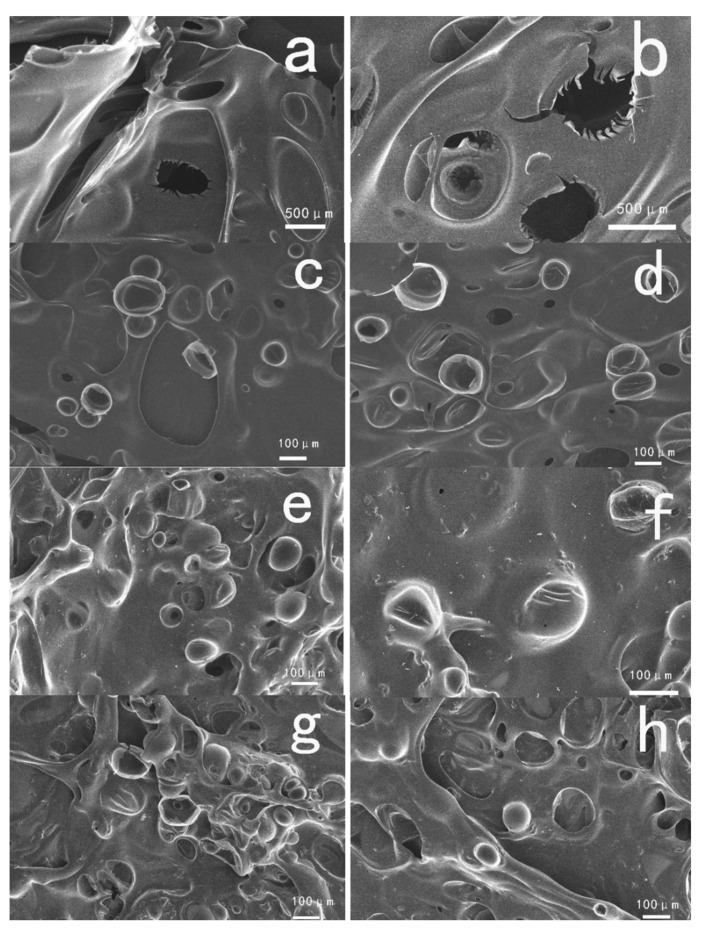
SEM image of PET, PET/1%CMSs-GDP, PET/2%CMSs-GDP, and PET/3%CMSs-GDP at 800 °C in a Muffle Furnace for 10 min under air atmosphere ((**a**,**b**) the outer surface of PET; (**c**,**d**) the outer surface of PET/1%CMSs-GDP; (**e**,**f**) the outer surface of PET/2%CMSs-GDP; (**g**,**h**) the outer surface of PET/3%CMSs-GDP).

**Figure 9 polymers-12-01689-f009:**
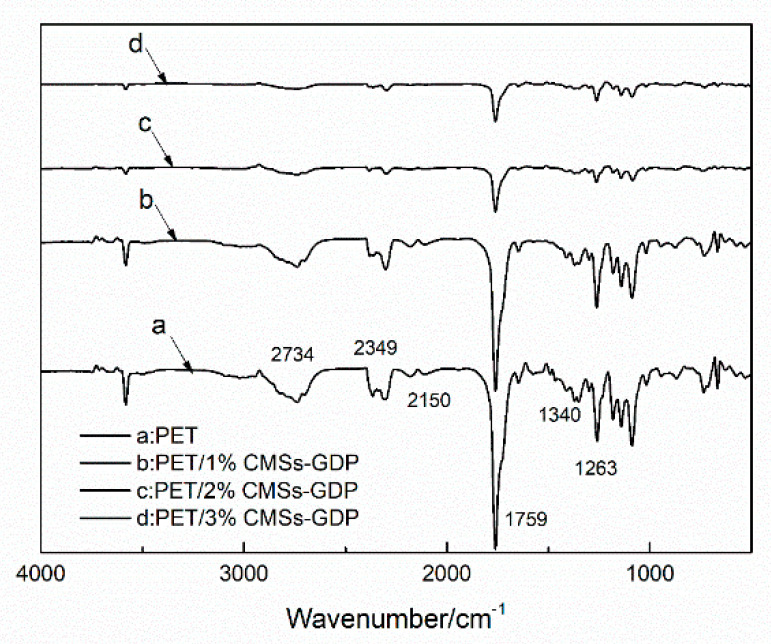
FTIR of pure PET and PET/CMSs composites at decomposition temperature (T_5%_).

**Figure 10 polymers-12-01689-f010:**
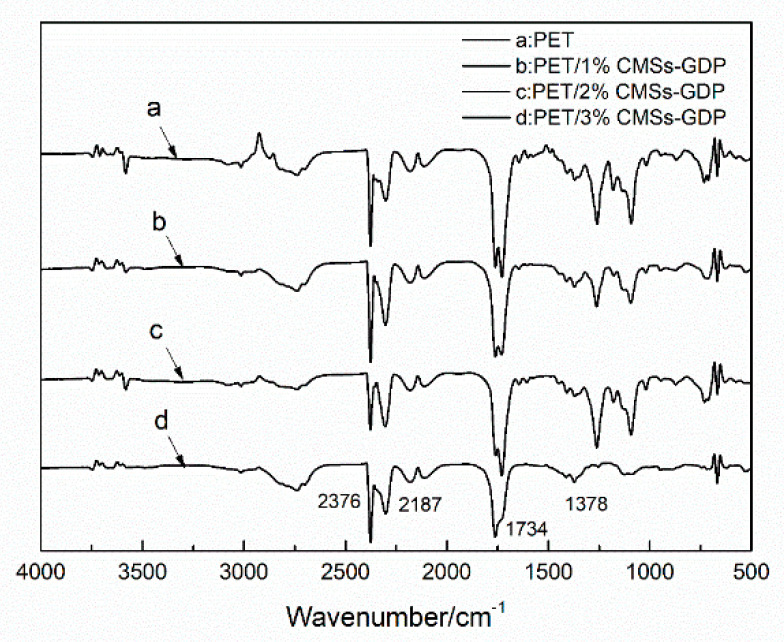
FTIR of PET composites at maximum decomposition temperature (T_max_).

**Figure 11 polymers-12-01689-f011:**
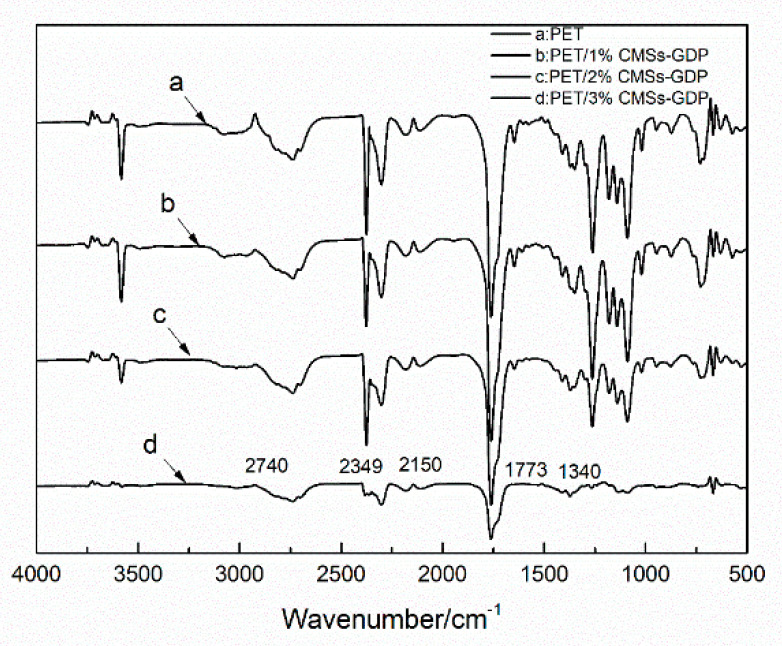
FTIR of PET composite at termination temperature (T_95%_).

**Figure 12 polymers-12-01689-f012:**
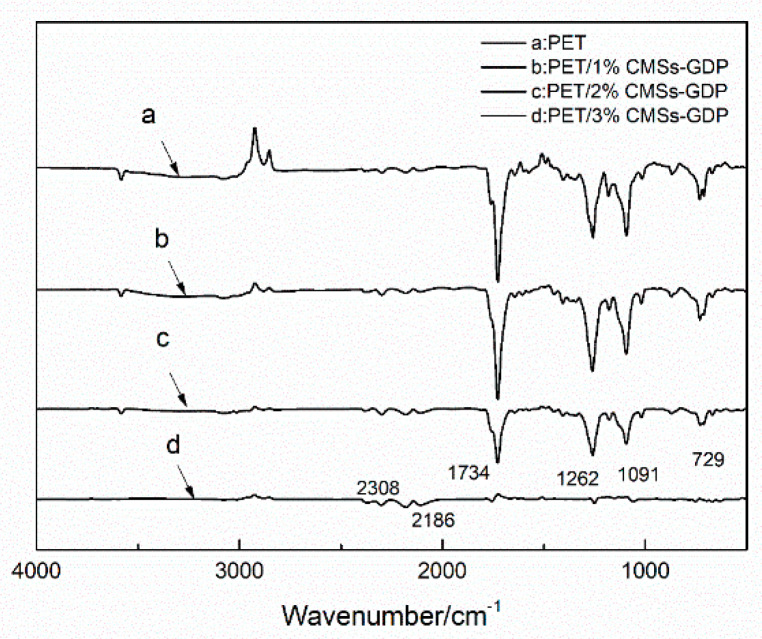
FTIR of PET composite at termination temperature (800 °C).

**Figure 13 polymers-12-01689-f013:**
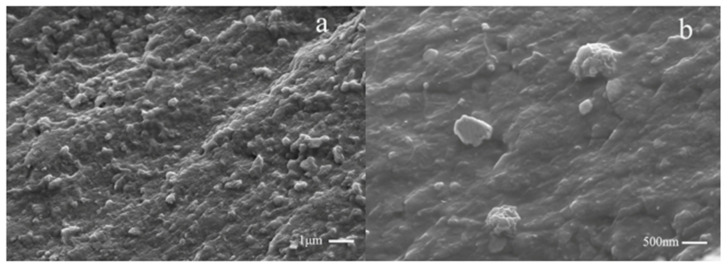
Scanning electron microscope image of PET composite material. (**a**) CMSs grafted with GDP; (**b**) CMSs with grafted GDP dispersed in the PET material.

**Figure 14 polymers-12-01689-f014:**
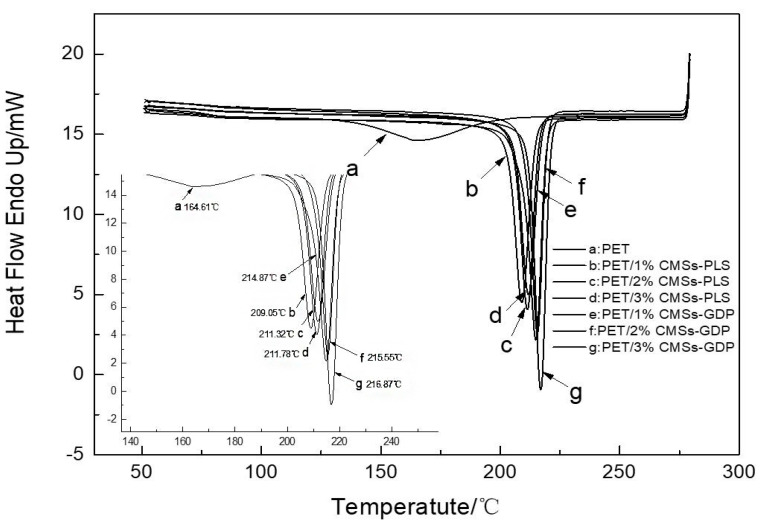
Cooling curve of the PET composite material.

**Figure 15 polymers-12-01689-f015:**
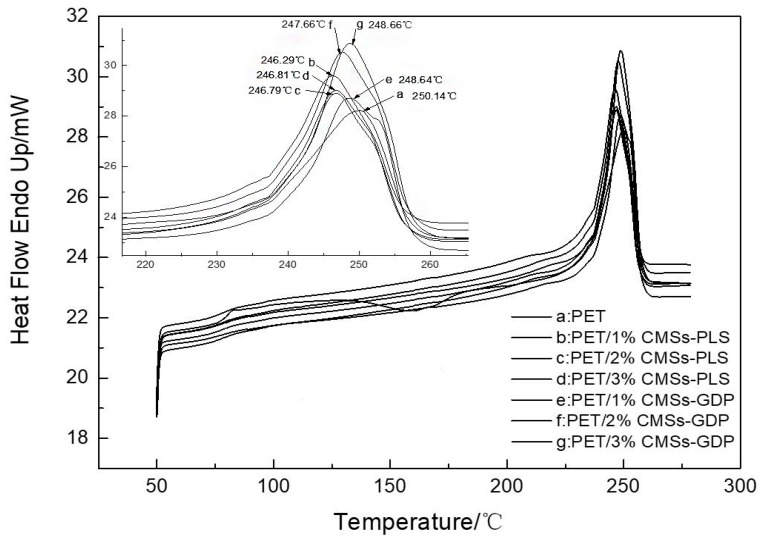
Heating curve of the PET composite material.

**Figure 16 polymers-12-01689-f016:**
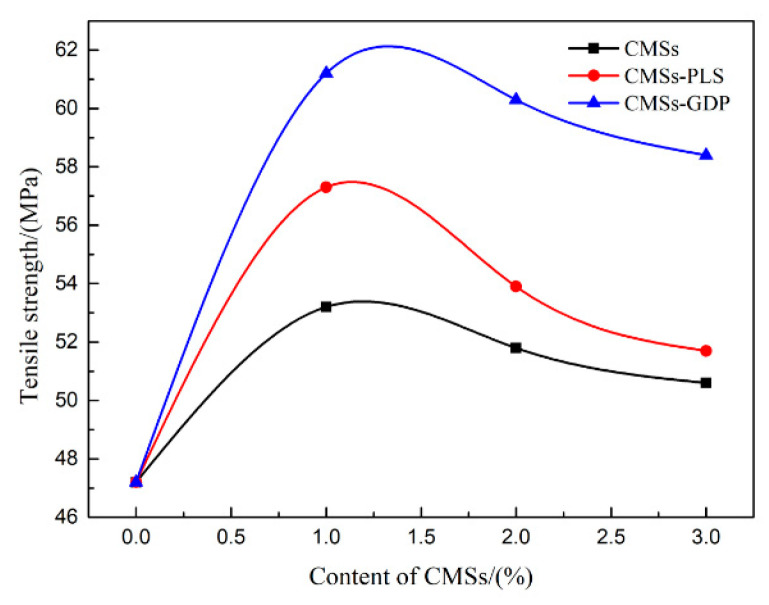
Tensile strength of composite materials.

**Figure 17 polymers-12-01689-f017:**
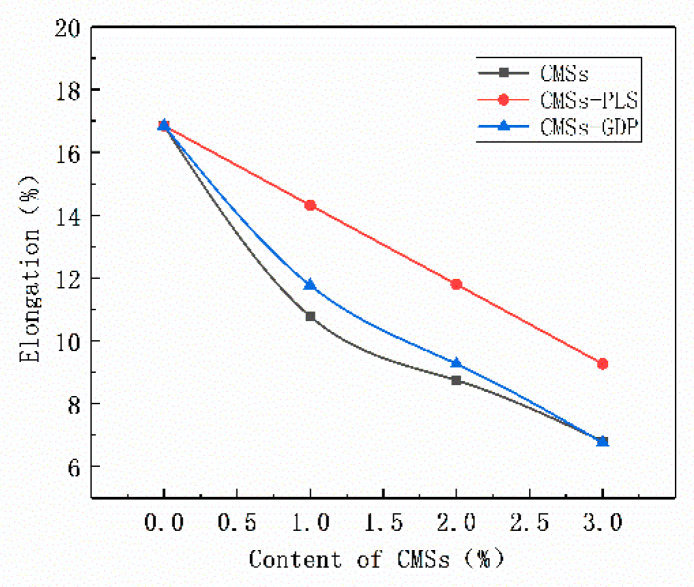
Elongation of composite materials.

**Table 1 polymers-12-01689-t001:** Grain size analysis of CMSs-PLS in ethyl alcohol solution.

Discharge Power/W	Time/min	Grain Size/nm	PDI
200	5	574	1.00
250	5	343	0.58
300	5	632	1.00
250	10	264	0.66
250	15	439	0.80

**Table 2 polymers-12-01689-t002:** Vertically burning analysis of polyethylene terephthalate (PET) composites.

Sample	t1/s	t2/s	t3/s	t1+t2/s	t2+t3/s	Melt Drip	Complete Combustion
100%PET	15.77	15.74	0.4	31.51	16.14	Y	Y
99%PET/1%CMSs	9.64	10.57	0.5	20.21	11.07	Y	Y
98%PET/2%CMSs	6.45	6.72	0.3	13.17	7.02	Y	Y
97%PET/3%CMSs	3.37	3.12	0.3	6.49	3.42	Y	N
99%PET/1%CMSs-PLS	9.34	7.78	0.6	17.12	8.38	Y	Y
98%PET/2%CMSs-PLS	4.21	5.35	0.5	9.56	5.85	Y	Y
97%PET/3%CMSs-PLS	3.21	3.44	0.2	6.65	3.46	Y	N
99%PET/1%CMSs-GDP	8.72	8.36	0.4	17.08	8.76	Y	Y
98%PET/2%CMSs-GDP	4.33	4.19	0.4	8.52	4.59	N	N
97%PET/3%CMSs-GDP	2.19	3.21	0.2	5.40	3.41	N	N

**Table 3 polymers-12-01689-t003:** TGA data of GDP, CMSs-GDP, PET, and the blends in nitrogen residue.

Sample	T_5%_ (°C)	T_max_ (°C)		The rate of T_max_ (wt % min^−1^)		Residue at 800 °C (wt %)
		Stage 1	Stage 2	Stage 1	Stage 2	
CMSs	352	-	440	-	10.8	55.31
CMSs-GDP	221	276	443	1.3	2.55	44.61
PET	420	-	464	-	39.8	11.91
PET/1%CMSs	422	-	464	-	38.2	13.20
PET/2%CMSs	423	-	466	-	37.5	13.21
PET/3%CMSs	426	-	468	-	36.4	13.78
PET/1%CMSs-GDP	428	-	470	-	35.9	14.18
PET/2%CMSs-GDP	431	-	472	-	35.1	14.65
PET/3%CMSs-GDP	435	-	473	-	34.4	15.42

**Table 4 polymers-12-01689-t004:** DSC data of the PET composite material.

Sample	Tm/°C	Tmc/°C	△Hm/(J/g)	△Hmc/(J/g)	Crystallinity/%
PET	250.14	164.61	38.90	26.06	22.27
PET/1%CMSs-PLS	246.29	209.05	43.16	45.48	30.82
PET/2%CMSs-PLS	246.79	211.32	44.80	46.32	32.00
PET/3%CMSs-PLS	246.81	211.78	43.35	47.33	30.96
PET/1%CMSs-GDP	248.64	214.87	51.00	49.98	36.43
PET/2%CMSs-GDP	247.66	215.55	47.55	50.46	33.96
PET/3%CMSs-GDP	248.66	216.87	48.00	52.29	34.29

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
