# Peer review of "Surface Modification of Carbon Microspheres with Guanidine Phosphate and Its Application as a Flame Retardant in PET"

_polymers, 2020, doi:10.3390/polym12081689_

Round 1
Reviewer 1 Report
Authors report here the synthesis of graft-modified carbon microspheres by the hydrothermal synthesis of cyclodextrin and further used for the delopment of PET based composites. The topic is interesting and the research is well planed. I only have some minor comments:
Introduction: (HNO3) should be (HNO3)
The scale of SEM images should be legible.
I suggest to perform all graphs with the same visual appearance. For instance, FTIR graph appearance's can be changed for that of the rest of graphs, TGA curves should be better in colours.
The tensile test results huld be extended and better discussed. For nstance, some comments regardng the elongation at break should be added. The results should be compared with similar materials already reported.
Author Response
尊敬的审稿人
非常感谢您对稿件的评论。根据您的建议,我们修改了稿件中的相关部分。下面回答了您的一些问题。
1)我们更正了文章中的书写错误(HNO3),在本文中以红色标记。
2)我们重新绘制了文章中SEM图像的比例,使其清晰可见;
3)我们根据您的意见对本文的FTIR图形进行了重新粉刷,使其外观与其余图形相同。我们还重新绘制了TGA曲线的颜色;
4)根据您的建议,我们增加了断裂伸长率的注释,并进行了深入的探索。
请查看附件了解详细信息

Reviewer 2 Report
Recently, carbon microspheres (CMSs) are attracting attention as flame-retardants for plastic materials.
The study of Jiang et al. is an interesting contribution to this novel field of research. The manuscript should be published in the journal polymers after a couple of supplementations and alterations (see below).
The authors synthesized CMSs from beta-cyclodextrin using a hydrothermal technique. The CMSs obtained this way were surface-treated with low-temperature plasma and subjected to a grafting reaction with guanidine phosphate. However, the Experimental section does not provide sufficient information to the second and third synthesis step. The authors should supplement section 2.1 (Material) by following details: Equipment and experimental conditions of surface-modification by low-temperature plasma, conditions applied to surface impregnation and grafting. In addition, the kind of reaction between guanidine phosphate and the reactive groups at the CMS-surface should be specified if possible (kind of reaction?).
The graft-modified CMSs were incorporated into polyethylene terephthalate (PET). Grain size analysis showed good dispersion of the graft-modified CMS particles in the PET matrix. Although low loading of the additive was applied (max. 3 wt%) good flame-retardant effect was achieved (V0 rating in UL 94 vertical burning test, pronounced increase of limiting oxygen index). The influence of the graft-modified CMSs on the thermal behavior of PET was investigated. Investigation to the flame-retardant action were performed by FTIR and char analysis by SEM. in case of CMS-containing PET samples, the SEM images revealed the formation of dense char that protects the underlying material.
Interestingly, the graft-modified CMSs were found to act as nucleating agent and forced the crystallization of PET. Moreover, an improvement of the tensile strength as a crucial material parameter was also obtained.
Due to these research findings the study can be of interest for the readers of polymers. However, the alterations mentioned above should be carried out. In addition, the manuscript has to be completed by insertion of keywords, address of corresponding author and acknowledgement. Please check the text again (there is a couple of missing blanks).
Author Response
Dear reviewer
I am very grateful to your comments for the manuscript.According with your advice,we amended the relevant part in manuscript. Some of your questions were answered below.
1)In the article, we have supplemented the equipment and experimental conditions of low-temperature plasma treatment, which have been marked in green in the article.
2)Your question about the type of reaction between the guanidine phosphate and the reactive groups on the surface of CMSs. After our in-depth investigation, we feel that the reaction between the guanidine phosphate and the reactive groups on the surface of CMSs should be an amidation reaction
